# Molecular Mechanism of Epidermal Barrier Dysfunction as Primary Abnormalities

**DOI:** 10.3390/ijms21041194

**Published:** 2020-02-11

**Authors:** Ai-Young Lee

**Affiliations:** Department of Dermatology, College of Medicine, Dongguk University Ilsan Hospital, 814 Siksa-dong, Ilsandong-gu, Goyang-si, Gyeonggi-do 410-773, Korea; lay5604@naver.com; Tel.: +8-23-1961-7250

**Keywords:** primary barrier dysfunction, epidermal calcium gradients, filaggrin, cornified envelopes, desquamation, skin lipids

## Abstract

Epidermal barrier integrity could be influenced by various factors involved in epidermal cell differentiation and proliferation, cell–cell adhesion, and skin lipids. Dysfunction of this barrier can cause skin disorders, including eczema. Inversely, eczema can also damage the epidermal barrier. These interactions through vicious cycles make the mechanism complicated in connection with other mechanisms, particularly immunologic responses. In this article, the molecular mechanisms concerning epidermal barrier abnormalities are reviewed in terms of the following categories: epidermal calcium gradients, filaggrin, cornified envelopes, desquamation, and skin lipids. Mechanisms linked to ichthyoses, atopic dermatitis without exacerbation or lesion, and early time of experimental irritation were included. On the other hand, the mechanism associated with epidermal barrier abnormalities resulting from preceding skin disorders was excluded. The molecular mechanism involved in epidermal barrier dysfunction has been mostly episodic. Some mechanisms have been identified in cultured cells or animal models. Nonetheless, research into the relationship between the causative molecules has been gradually increasing. Further evidence-based systematic data of target molecules and their interactions would probably be helpful for a better understanding of the molecular mechanism underlying the dysfunction of the epidermal barrier.

## 1. Introduction

The skin functions as a barrier against the environment by protecting from mechanical insults, microorganisms, chemicals, and allergens. Tight junctions contribute to the formation of the skin barrier in the granular cell layer. However, the stratum corneum (SC), the outermost layer of skin, plays a main role in the formation of the skin barrier. The SC consists of several layers of corneocytes with cornified envelopes (CEs), corneodesmosomes, and intercellular lipid lamellae. 

Structural and functional impairment of the epidermal barrier can allow irritants and allergens to penetrate the SC, which could lead to various skin diseases, including atopic dermatitis, irritant contact dermatitis, and allergic contact dermatitis [1,2,3,4]. The role of epidermal barrier disruption in development and progression of these skin disorders has been demonstrated based on clinical findings and/or non-invasive parameters for skin irritation evaluation. The parameters include transepidermal water loss, electrical conductance, surface pH, lipid composition, skin blood flow, skin color, and skin thickness [5,6]. However, these skin diseases could reversely damage the skin barrier, thereby resulting in a vicious cycle [7]. It may be difficult to specify which portions of these diseases, particularly contact dermatitis (either irritant or allergic), are the result of barrier disruption, not the cause. To determine the role of the epidermal barrier in development and progression of skin disorders, barrier abnormalities should be developed as primary events. Ichthyoses caused by monogenic defects are representative skin diseases associated with primary barrier dysfunction [8]. In atopic dermatitis, barrier abnormalities may not be the primary events in lesional skin. However, they can play a role in part as primary abnormalities because abnormalities in the epidermal barrier are already present in non-lesional skin of atopic dermatitis [1]. Epidermal trauma from tape stripping or irritant application also leads to initial disruption of the barrier, although the result at certain time points may be combined with compensatory reactions followed by trauma.

Causative mechanisms involved in primary epidermal barrier dysfunction might be more valuable for the prevention of skin diseases induced by barrier abnormalities. However, the underlying mechanism has been discussed mainly in connection with immunologic responses, including cytokine production [1,9,10]. Accordingly, this review covers molecular mechanisms of epidermal barrier dysfunction as primary abnormalities by focusing on the causative factors of skin diseases (atopic dermatitis and ichthyoses) and experimental skin conditions (tape stripping and irritant application) associated with skin barrier abnormalities.

## 2. Molecular Mechanisms Related to Epidermal Barrier Dysfunction

Formation and maintenance of skin barrier integrity could be influenced by genetic and environmental factors involved in epidermal cell differentiation and proliferation, cell–cell adhesion, and skin lipids. Mechanisms related to epidermal barrier dysfunction at molecular levels have commonly illustrated filaggrin mutations [3,11]. Regarding the mechanism behind skin barrier integrity, the role of epidermal calcium gradients could be considered based on their influence on keratinocyte proliferation and differentiation [12]. The roles of the structural components of the epidermal barrier such as CEs and skin lipids in maintaining epidermal barrier integrity should be considered. In addition, factors related to dynamic equilibrium, including desquamation, is probably involved in maintaining epidermal barrier integrity.

### 2.1. Epidermal Calcium Gradients

The mammalian epidermis shows a characteristic calcium gradient formed mainly by Ca^2+^ release from endoplasmic reticulum stores and Ca^2+^ influx from extracellular sources. Calcium gradients are not confined to SC, but across the epidermis. Calcium levels are low in basal and spinous layers, whereas the levels are increasing up to the granular layers. Calcium levels decline again in the SC. The epidermal calcium gradient plays a crucial role in keratinocyte differentiation and epidermal barrier formation. On the other hand, the gradient disappears after acute barrier disruption and reforms with barrier function recovery, indicating that the epidermal barrier can inversely regulate the formation of the calcium gradient [13]. Therefore, it may not be easy to determine whether the calcium gradient disappearance or barrier disruption is the primary event under certain conditions.

The association between epidermal calcium gradients and epidermal barrier integrity and function can be inferred by the keratitis-ichthyosis-deafness (KID) syndrome. As the name of the syndrome suggests, ichthyosis is a main skin abnormality accompanied by palmoplantar keratoderma and erythrokeratoderma. KID syndrome could be caused by heterozygous dominant missense mutations in GJB2 (gap junction protein beta 2) and GJB6 (gap junction protein beta 6) genes, encoding connexin 26 (Cx26) and Cx30, respectively. These mutations increase the release of ATP and Ca^2+^ influx, which disturbs the epidermal calcium gradient leading to a barrier defect [14]. Hyperkeratosis such as palmoplantar keratoderma and erythrokeratoderma in KID syndrome has been considered as a compensatory reaction to the barrier defect. Results resembling human KID syndrome have been exhibited by mice with heterozygous mutations of Cx26 (Cx26S17F), which supports the association between altered epidermal calcium gradients and defective epidermal barriers [15]. In addition to these pathologic conditions, aging skin has been associated with a decreased epidermal barrier function. The turnover rate of keratinocytes slows down with aging. A role for calcium in skin aging has been demonstrated based on the epidermal calcium gradient collapse during the aging process [16].

As for the formation and homeostasis of the epidermal calcium gradient in keratinocytes, roles of calcium-sensing receptor (CaR), epidermal calcium channels (transient receptor potential (TRP) channels, store-operated calcium entry (SOCE) channels, and voltage-gated calcium channels), and calmodulin-like skin protein (CLSP) have been addressed [17,18,19]. The role of CaR in epidermal barrier homeostasis could be elucidated by the results obtained from keratinocyte-specific CaR knockout mice [20]. A defective epidermal barrier displayed in the knockout mice and abnormal Ca^2+^ influx with decreased differentiation in the keratinocytes cultured from these mice indicates that epidermal calcium gradients altered by CaR deletion impair keratinocyte differentiation and epidermal barriers. Growing evidence indicates that TRP channels can regulate skin barrier homeostasis with keratinocyte differentiation and proliferation. TRP channels respond to changes in environmental factors, such as activation of TRP vanilloid type 1 (TRPV1) by heat (42 °C), capsaicin or TRPV4 by heat (>30 °C), and hypo-osmolarity [21]. In the case of TRPV1, blockade of activation using a TRPV1 antagonist compound can suppress atopic dermatitis-like symptoms by accelerating skin barrier recovery in atopic dermatitis murine models [22]. On the other hand, the blockade of TRPV4 activation by physical and chemical stimuli has been reported to possibly disrupt epidermal barrier integrity and homeostasis in human keratinocytes, ex vivo human skin, and TRPV4-null mice [23]. Two proteins, STIM1 (stromal interaction molecule1) and Orai1 (ORAI calcium release-activated calcium modulator 1), have been identified as essential components of SOCE in human keratinocytes. STIM1 as a calcium sensor of endoplasmic reticulum can activate Orai1 when endoplasmic reticulum calcium levels decrease. Orai1 downregulation can abolish the calcium-switch-induced calcium response, resulting in impaired keratinocyte differentiation and epidermal barrier in human keratinocytes and in Orai1-knockout mice [24]. Orai1 activation can also cause atopic dermatitis with barrier dysfunction. However, the mechanism involves that Orai1 activation has induced Th2- and Th22-deviated immune reactions due to the release of a large amount of TSLP (thymic stromal lymphoprotein) from keratinocytes [25]. A new calcium-binding protein, CSLP, is particularly abundant in differentiated epidermises. It can modulate calcium-dependent proteins involved in epidermal barrier formation [18,26]. Although CLSP has been related to atopic dermatitis, the upregulation in the epidermis of acutely exacerbated but not non-exacerbated atopic dermatitis [18] suggests that CSLP upregulation in atopic dermatitis is the compensatory reaction for barrier homeostasis.

Calcium is a prerequisite for keratinocyte differentiation. In addition, it has been demonstrated that calcium gradients could regulate skin lipid compositions through the formation and secretion of lamellar bodies as revealed by a marked decrease in linoleoyl ω-esterified ceramides in the KID syndrome [14,15]. Calcium gradients are also involved in CE rearrangement through the synthesis of CE components and cross-linking to the plasma membrane, as shown by chronological skin aging [17]. These findings indicate that epidermal calcium gradients may perhaps play a role in cooperation with other mechanisms involved in epidermal barrier homeostasis.

Collectively, the findings identified in KID syndrome and chronological aging provide evidence for the role of epidermal calcium gradients in epidermal barrier function. The data from experimental skin conditions also suggest a potential role of molecules involved in epidermal calcium gradient formation (CaR, TRPV1, TRPV4, and Orai1) in epidermal barrier homeostasis (Figure 1), although their clinical significance remains to be clarified.

### 2.2. Filaggrin and Cornified Envelopes

Filaggrin is produced as a polymer profilaggrin. Filaggrin as the major keratin-binding protein is stored in keratohyaline granules in the form of the keratin–profilaggrin complex in the granular layer. During cornification, profilaggrin is cleaved into filaggrin monomers by several proteases. SASPase (skin aspartic protease) is a candidate protease of profilaggrin linker-cleavage [27]. In addition to binding to keratin filaments inside corneocytes, filaggrin also cross links to CEs [28].

The best known molecular mechanisms related to defective barrier function are mutations in filaggrin (Figure 2). In addition to cross-linking to CE, filaggrin monomers degrade to form urocanic acid and pyrrolidine carboxylic acid, which contribute to an acidic skin pH and water-holding capacity. Skin pH influences multiple factors involved in epidermal barrier integrity, such as lipid synthesis and desquamation by the regulation of enzymatic activity [29,30]. It has been demonstrated that a chain of these filaggrin functions are involved in defective barrier function induced by filaggrin deficiency. Loss-of-function mutations in *FLG*, which encodes filaggrin, have been verified in atopic dermatitis [28,31]. FLG mutations have also been associated with other skin disorders, such as ichthyosis vulgaris [11], occupational contact dermatitis [3,32], and chronic hand eczema [33]. Filagrin monomers are formed by SASPase. Loss-of-function mutations in *Asprv1* encoding aspartic peptidase, retroviral-like 1, which is also known as SASPase, cannot form filaggrin thereby leading to a filaggrin deficiency with dry skin in mice [34]. Although the finding suggests the role of filaggrin metabolic process in atopic dermatitis, ASPRV1 mutations are not associated with atopic dermatitis or dry skin in humans [35]. Instead, ASPRV1 mutations are associated with ichthyosis in dogs [36].

Well-established mechanisms of the action of filaggrin on the maintenance of structural and functional epidermal barrier homeostasis may contribute to the gradual expansion of the role of filaggrin in skin disorders other than atopic dermatitis (Figure 2). Interest in research on the filaggrin metabolic process has also been increasing.

### 2.3. Cornified Envelopes

CEs are the most insoluble components formed beneath the plasma membrane of corneocytes. CEs are composed of various molecules, such as involucrin, loricrin, small proline-rich proteins (SPRRs), envoplakin, periplakin, and cysteine protease inhibitor A (cystatin A). The molecules are crossed-linked by transglutaminases (TGases).

Associations between mutations in CE precursor SPRR and SPRR3 genes and atopic dermatitis have been reported [37,38]. Mutations in loricrin, a major component of CE making up to 70% of the SC protein, are most frequently associated with loricrin keratoderma. Downregulation of loricrin is also frequently associated with atopic dermatitis [39]. However, only mild symptoms develop with normal-looking CEs in loricrin knock-out mice [40]. Envoplakin and periplakin play a role in linking them to intermediate filaments [41]. However, triple knocking-out of envoplakin, periplakin, and involucrin is required to induce abnormal CEs [42]. Mutations or deficiencies of involucrin, despite being a major component of CE, may not be enough for the development of atopic dermatitis. Only a subtle phenotype by mutations or deficiencies of these major CE components may indicate the existence of strong compensatory mechanisms. Loss-of-function mutation in *CSTA*, a gene encoding cystatin A, can induce autosomal recessive exfoliative ichthyosis with reduced thickness of CE and abnormalities in the lamellar body [43]. Acral peeling skin syndrome, which is an autosomal recessive genodermatosis, characterized by asymptomatic peeling of the hands and feet, can be caused by the loss-of-function mutation in *CSTA* as well [44].

Acral peeling skin syndrome can also be induced by missense mutations in TGase5 [45]. TGase1 deficiency from a mutation in *TGM1* (transglutaminase 1) encoding the TGase1 enzyme can cause lamellar ichthyosis, an autosomal recessive congenital ichthyosis [46,47]. The role of Tgase1 or Tgase5 in epidermal barrier function based on CE formation has also been identified in mice. However, the disruption of the CEs induced by the loss of Tgm1 expression in embryos does not develop in adults due to compensatory upregulation of Tgm5 expression. For perturbation of the CEs, loss of both Tgm1 and Tgm5 expression is considered essential in adult mice [48]. Although it is unclear whether different effects of TGase deficiencies on CE disruption could be dependent on the time required to develop the deficiency, their genetic variations have not been associated with atopic dermatitis. Instead, a significant increase in TGM3 mRNA expression has been observed in atopic dermatitis [49].

Reduced CE components can also be induced by prevailing experimental conditions of epidermal barrier disruption. Retinoic acid has been frequently used to induce dryness and scaling in the applied skin [50]. It can reduce levels of loricrin and SPRRs in human keratinocytes and mice skin in a dose-dependent manner [51]. Different from these results on atopic dermatitis, ichthyoses, and retinoic acid applications, expression levels of several CE proteins have been found to be increased in epidermal barrier disruption of human skin by tape stripping and in skin irritation by sodium dodecyl sulfate application [52]. However, such findings are considered as a compensatory reaction for barrier repair.

As reviewed here, the evidence-based or potential association of several CE components with skin disorders and experimental conditions showing barrier disruption indicates the role of CE components in barrier homeostasis (Figure 3). However, minimal changes are induced by ablation of specific single CE structural components.

### 2.4. Desquamation

Desquamation is the gradual invisible shedding of corneocytes, which is determined by the de novo synthesis and degradation of corneodesmosomal proteins. Corneodesmosomes are modified desmosomes formed while keratinocytes differentiate from granular cell layers into cornified cell layers. They play a critical role in cell–cell adhesion of corneocytes. Corneodesmosomes are constituted with desmoglein 1, desmocollin 1, and corneodesmosin. Degradation of corneodesmosomal proteins is controlled by proteases and a variety of inhibitors. Kallikrein-related peptidases (KLKs) and cathepsins are included in proteases. Protease inhibitors, cholesterol sulfate, and an acidic gradient are considered as inhibitors of degradation. Fifteen different KLK family serine proteases have been detected in normal human skin [53]. The activity of the serine proteases of the KLKs is controlled by serine protease inhibitors, including anti-leukoprotease, elafin (skin-derived anti-leukoprotease; SKALP), and lymphoepithelial-Kazal-type 5 inhibitor (LEKTI) [54,55].

Defects in corneodesmosome components themselves have been involved in corneodesmosome abnormality and lead to epidermal barrier impairment as follows: Abnormalities in genes encoding corneodesmosin can cause a generalized inflammatory type of peeling skin syndrome and those encoding desmoglein 1 can induce another generalized inflammatory type of peeling skin syndrome, SAM syndrome (severe skin dermatitis, multiple allergies, and metabolic wasting) [56,57].

Corneodesmosome degradation is accelerated whenever protease activities overcome activities of protease inhibitors, resulting in the premature breakdown of corneodesmosomes. One of the well-known examples is the severe autosomal recessive form of ichthyosis, Netherton syndrome, caused by a defect in the serine-specific inhibitor Kazal type 5 (SPINK5) gene encoding LEKTI. Netherton syndrome has also been proposed as a generalized inflammatory type of peeling skin syndrome [56]. Lack of serine protease inhibition can increase activities of KLKs, thereby causing acceleration in the degradation of the corneodesmosome [57,58]. Combined reduction of proteolytic activity by KLK5 downregulation in the presence of loss-of-function SPINK5 mutation in an animal model or an organotypic skin culture model has reversed the symptoms with restoration of corneodesmosome structure and severe skin barrier defects [59,60]. These findings emphasize that corneodesmosome degradation depends on the sum of protease and inhibitor activities. In addition to increased activity of serine proteases, TGase1 activity has been reported to increase in Netherton syndrome, which contributes to the novel functional link between LEKTI and TGase1 [61]. Considering that impaired corneodesmosome degradation can lead to hyperkeratosis as a clinical symptom, hyperkeratosis is not considered as a cause of barrier dysfunction related to desquamation. In fact, hyperkeratosis occurs in lesional skin of atopic dermatitis due to an increase in LEKTI as a compensatory reaction to overcome upregulated KLK7 activities with reduced overall proteolytic activity [62]. However, hyperkeratosis also develops in non-lesional skin of atopic dermatitis due to reduction in serine protease activity, different from the mechanism of lesional skin of atopic dermatitis [63]. Similarly, in autosomal recessive ichthyosis-hypotrichosis syndrome caused by a loss-of-function mutation of the ST14 gene encoding matriptase, a type II transmembrane serine protease, impairment of corneodesmosome degradation has been reported [64,65]. Because barrier abnormalities in non-lesional skin of atopic dermatitis and ichthyoses caused by monogenic defects are considered as primary events [1,8], desquamation might not be a mechanism involved in barrier dysfunction in atopic dermatitis and a certain type of ichthyoses.

Experimental conditions could affect skin barrier impairment via an imbalance in activities between proteases and inhibitors. Retinoic acid may possibly facilitate penetration of applied agents by loss of corneocyte cohesion [66]. Increased proteolytic activity from KLK upregulation without a change in LEKTI has been identified as a mechanism of retinoic acid-induced accelerated desquamation [53]. Washing with soap or detergents and long-term application of topical corticosteroids could increase KLK production [67].

Collectively, accelerated desquamation either by reduced corneodesmosome synthesis or increased corneodesmosome degradation is linked to skin disorders and barrier dysfunction under experimental conditions (Figure 4), suggesting an important role of desquamation in epidermal barrier integrity and homeostasis.

### 2.5. Skin Lipids

Skin lipids that constitute the extracellular matrix of the SC are composed of cholesterol, free fatty acids, and ceramides. These lipids are stacked to form densely packed lipid layers, lipid lamellae, which depends on the composition of the lipids. Precursor lipids are synthesized in keratinocytes, although sebaceous gland-derived lipids and extracutaneous sources could contribute to the epidermal lipid pool [68,69]. SC lipids are secreted from keratinocytes into extracellular space via lamellar bodies, which contain phosphoglycerides, sphingomyelin, and glucosylceramides. After secretion, these lipids are further metabolized by enzymes co-secreted in lamellar bodies, which include glucocerebrosidase, sphingomyelinase, and phospholipase A [68]. Peroxisome proliferating activated receptor (PPAR) isoforms (alpha, beta/delta, and gamma) and liver X receptor (LXR) isoforms are expressed in the epidermis. Activation of these receptors can stimulate epidermal lipid metabolisms, such as epidermal lipid synthesis, lamellar body formation and secretion, and activity of enzymes involved in extracellular processing of lipids in the SC [70].

Changes in lipid composition induce abnormal lipid organization leading to impaired epidermal barrier function [71]. Most of the reports are related to abnormalities in ceramide synthesis. Ceramide is a structural backbone of sphingolipids and demonstrates structural diversity particularly in the epidermis. Changes in ceramides, including levels, composition, and chain-lengths, are the most distinctive hallmark of atopic dermatitis [72,73]. An open-label clinical study [74] has shown that application of ceramide-dominant lipid mixture can improve atopic dermatitis symptoms with a decrease in transepidermal water loss, supporting the important role of ceramide in atopic dermatitis. Sphingomyelin in the epidermis is a precursor of ceramide. Sphingomyelin synthase (SGMS) generates sphingomyelin. Sphingomyelin and ceramide contents have been decreased in Sgms2-knockout mice with barrier dysfunction [75]. Ceramides are derived from glucosylceramides, which are synthesized by UDP-glucose ceramide glucosyltransferase (UGCG). UGCG deficiency induces depletion of glucosylceramides and ichthyosis-like skin phenotype in mice [76]. Abnormal synthesis of ceramides, particularly ultra-long-chain acylceramide, exists in the epidermis and has been involved in the pathogenesis of various ichthyoses and ichthyosis syndromes. Loss-of-function mutations in NIPAL4 (NIPA like domain containing 4) cause autosomal recessive congenital ichthyosis. Acylceramide levels are reduced with impaired lipid multilayer structure formation in Nipal4-knockout mice, which indicates the role of NIPAL4 deficiency in epidermal barrier defects [77]. CYP4F22 is a fatty acid ω-hydroxylase involved in the synthesis of acylceramide. Loss-of-function mutations in CYP4F22 cause autosomal recessive congenital ichthyosis. Impaired lipid lamella formation with almost complete loss of acylceramide and its precursor ω-hydroxyceramide have been demonstrated in knockout mice [78]. Loss-of-function mutations in ALOX12B and ALOXE3, which are essential for the generation of ω-hydroxyceramide, can cause autosomal recessive congenital ichthyosis [79,80]. Patatin-like phospholipase domain-containing lipase 1 (PNPLA1) is involved in the biosynthesis of ω-О-acylceramide, a particular class of sphingolipids, by catalyzing the ω-О-esterification of linoleic acid [81]. PNPLA1 gene mutations also cause autosomal recessive congenital ichthyosis by a blockade in the production of ω-О-acylceramide with a concomitant accumulation of their precursors [82,83,84]. α/β hydrolase domain-containing protein 5 (ABHD5) has been identified to interact with PNPLA1 as a coactivator. It could be easily inferred that ABHD5 mutations can decrease epidermal ω-О-acylceramide synthesis, thereby causing ichthyosis [85]. Comparative gene identification-58 (CGI58) deficiency also impairs ω-О-acylceramide synthesis with severe barrier defects in Cgi58-deficient mice [86]. Mutations in *SDR9C7*, encoding a short-chain dehydrogenase/reductase family 9C member 7 (SDR9C7), have been recently found in ichthyosis. Complete loss of a species of ω-О-acylceramide esterified with linoleate-9,10-trans-epoxy-11E-13-ketone has been proposed as the mechanism based on the result from a knockout Sdr9c7 mouse [87]. Long-term use of topical glucocorticoids could alter the SC lipid profiles, particularly long-chain ester-linked ceramide, thereby leading to skin barrier defects [88].

Despite the fact that free fatty acids and cholesterol are the major constitutes of skin lipids, not much has been reported about the association between altered synthesis of these lipids and skin barrier dysfunction. Fatty acid elongases (ELOVL) are involved in the synthesis of very long-chain fatty acids. A distinct set of ELOVL4 mutations can cause a neurocutaneous disorder characterized by ichthyosis, seizures, spasticity, intellectual disability, and ichthyosis [89]. Inhibition of cholesterol synthesis possibly due to loss-of-function mutations in the NADP dependent steroid dehydrogenase-like (NSDHL) gene can cause CHILD syndrome (Congenital Hemidysplasia with Ichthyosiform Erythroderma and Limb Defects) [90]. However, cholesterol depletion, which reduces the content of incorporated cholesterol to approximately half of the normal value, has shown a negligible effect on the lipid chain order without compromising the barrier function in isolated human SC [91]. Under the conditions of grouping of depletion with Th2 cytokines, cholesterol depletion probably induces atopic dermatitis-like alteration with epidermal barrier impairment [92].

Concerning lamellar body formation and secretion, some evidence-based data have been reported. Ceramides are derived from glucosylceramides after the secretion of lamellar bodies. ATP-binding cassette transporter A12 (ABCA12) can facilitate the delivery of glucosylceramides to lamellar bodies in keratinocytes. A loss-of-function mutation in ABCA12 impairs lipid lamellar membrane formation in the SC, causing harlequin ichthyosis, which is the most severe phenotype of autosomal recessive congenital ichthyosis [93,94]. Although ABCA12 deficiency has not been identified in atopic dermatitis, their association may be deduced from the result showing that ceramides can upregulate ABCA12 expression via the PPAR-mediated signaling pathway [95]. Mutations in VPS33B and VIPAS39 cause ARS syndrome (arthrogryposis, renal dysfunction, and cholestasis). Mice knockout either of VPS33B or VIPAS39 have shown abnormal morphology and localization of lamellar body with reduced thickness of CEs and deposition of lipids in the SC [96]. Loss-of-function mutation in *CSTA* as described in Section 2.5 (Cornified Envelopes) has displayed premature secretion of the lamellar body and a delayed processing of the secreted lamella body contents [43]. The result from Rab11a silencing in reconstituted human epidermis suggests that Rab11a GTPase could regulate lamellar body biogenesis [97]. Deficiency of fatty acid transport protein 4 (FATP4) can cause ichthyosis prematurity syndrome. FATP4 is an acyl-CoA synthetase, one of the proteins involved in the uptake of long-chain fatty acids by regulating fatty acyl moieties. Abnormalities in the lamellar body are detected in Fatp4-knockout mice with an altered skin lipid composition [98]. Although alteration in lipid transporters has been mostly associated with ichthyoses, FATP4 mutation has also induced atopic dermatitis [31].

These evidence-based data indicate that most of the skin disorders showing skin lipid defects have been reported in connection with abnormalities in lipid composition. There have been only a few reports on lamellar body formation and secretion, or activity of enzymes involved in extracellular processing of lipids in the SC (Figure 5a,b).

## 3. Conclusions

This review shows that abnormalities in factors involved in epidermal barrier integrity, such as epidermal calcium gradients, filaggrin, cornified envelopes, desquamation of corneodesmosomes, and skin lipids, are associated with epidermal barrier dysfunction. The data identified in skin disorders caused by monogenic defects related to epidermal barrier dysfunction could provide reliable clues or insight into the underlying mechanisms concerning epidermal barrier dysfunction as primary events. Factors involved in ichthyoses, atopic dermatitis without exacerbation or lesion, and early time of experimental irritation could be considered as causes and not results of skin barrier dysfunction. A critical role of filaggrin and ceramides in skin barrier function has been elucidated from investigations on ichthyoses, atopic dermatitis, and experimental conditions. Based on clinical and experimental data, causative roles of a few molecules involved in epidermal calcium gradients formation and homeostasis, some components of CEs, KLKs/LEKTI, and molecules involved in lipids synthesis/transport in barrier homeostasis have been identified.

Under certain conditions, more than one abnormal finding is present. Examples include abnormal lipid composition of the SC associated with impaired epidermal calcium gradients in KID syndrome [14,15], altered calcium gradients accompanied by increased skin pH and CE rearrangement in aging skin [16], loss-of-function mutations in filaggrin in atopic dermatitis [28], and increased activity of TG1 and serine proteases in Netherton syndrome [61]. These accompanying findings could provide a direction for future research to identify more organized mechanism involved in barrier dysfunction.

Most of these findings were episodic and some data were obtained from experimental conditions. However, an increasing number of researches is being carried out on the molecular mechanisms of target molecules identified based on clinical conditions. These approaches could be helpful to get evidence-based systematic data on the exact mechanism of epidermal barrier dysfunction.

## Figures and Tables

**Figure 1 ijms-21-01194-f001:**
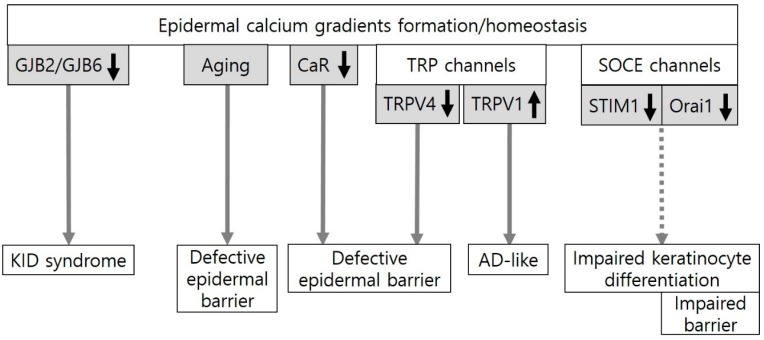
An evidence-based or potential association between epidermal calcium gradients and the epidermal barrier. Epidermal calcium gradients have been altered in keratitis-ichthyosis-deafness (KID) syndrome induced by gap junction protein beta 2 (GJB2) or GJB6 missense mutations and in chronological skin aging. The formation and homeostasis of the epidermal calcium gradients could be regulated by the calcium-sensing receptor (CaR), transient receptor potential (TRP) channels, and store-operated calcium entry (SOCE) channels. CaR deletion, which inhibits Ca^2+^ influx, impairs the epidermal barrier in mice. In TRP channels, TRP vanilloid 4 (TRPV4) activation, which could be induced by heat (>30 °C) and hypo-osmolarity, plays an important role in epidermal barrier formation and recovery in mice. On the other hand, the blockade of TRPV1 activation by physical and chemical stimuli such as heat (42 °C) and capsaicin can suppress atopic dermatitis (AD)-like symptoms in mice. Two essential components of SOCE, STIM1 (stromal interaction molecule1) and Orai1 (ORAI calcium release-activated calcium modulator 1), are activated by endoplasmic reticulum (ER) calcium store depletion. Downregulation of Orai1 can impair keratinocyte differentiation and barrier homeostasis in mice.

**Figure 2 ijms-21-01194-f002:**
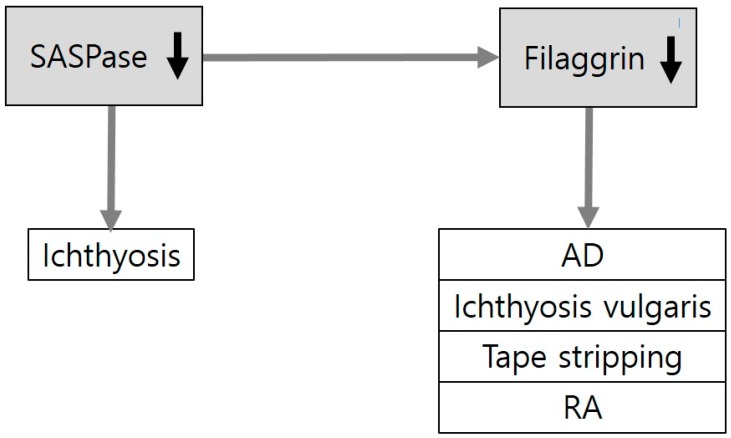
Downregulation of filaggrin in skin diseases and experimental conditions related to epidermal barrier dysfunction. Loss-of-function mutations of FLG (encoding filaggrin) are associated with skin disorders related to barrier dysfunction, including atopic dermatitis (AD) and ichthyoses. Expression levels of filaggrin have also been reported to be reduced in experimental conditions of the disrupted epidermal barrier by tape stripping or retinoic acid (RA) application. SASPase (skin aspartic protease) generates filaggrin monomers from profilaggrin. Loss-of-function mutations of SASPase are associated with ichthyosis in dogs.

**Figure 3 ijms-21-01194-f003:**
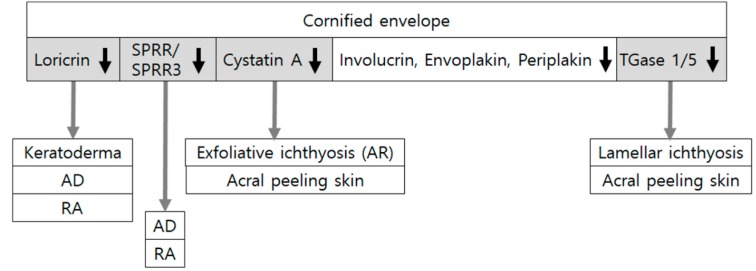
Downregulation of cornified envelope components in skin diseases and experimental conditions related to epidermal barrier dysfunction. Cornified envelopes are composed of various molecules, such as involucrin, loricrin, small proline-rich proteins (SPRRs), envoplakin, periplakin, and cysteine protease inhibitor A (cystatin A), crossed-linked by transglutaminases (TGases). Loss-of-function mutations in SPRR/SPRR3, LOR (encoding loricrin), TGM1/TGM5 (encoding transglutaminase1/5), and CSTA (encoding cystatin A) have been associated with atopic dermatitis (AD) or ichthyoses. Expression levels of loricrin and SPRRs have been reduced in experimental conditions of the disrupted epidermal barrier by retinoic acid (RA) application. Although triple knocking-out of envoplakin, periplakin, and involucrin induces abnormal cornified envelopes, mutations or deficiencies of involucrin alone do not cause atopic dermatitis.

**Figure 4 ijms-21-01194-f004:**
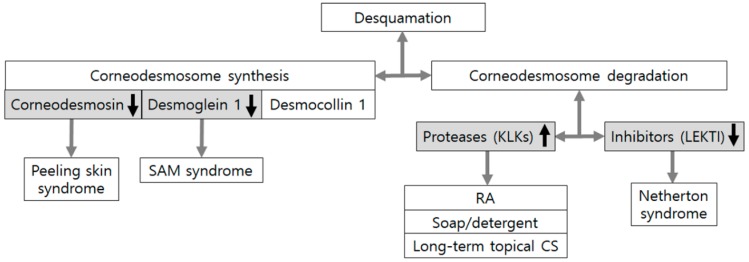
Corneodesmosome defects in skin conditions associated with epidermal barrier dysfunction. Desquamation is determined by de novo synthesis and degradation of corneodesmosomal proteins. Mutations in *GDSN* encoding corneodesmosin and *DSG1* encoding desmoglein 1 can cause an inflammatory type of peeling skin syndrome and SAM syndrome (severe skin dermatitis, multiple allergies, and metabolic wasting), which is another inflammatory type of peeling skin syndrome, respectively. Degradation of corneodesmosomal proteins depends on the sum of activities from proteases, including kallikrein-related peptidases (KLKs) and protease inhibitors including lymphoepithelial-Kazal-type 5 inhibitor (LEKTI). Experimental conditions, such as retinoic acid (RA) application, soap and detergent washing, and long-term corticosteroid (CS) application, could also accelerate desquamation mainly by increased production of KLKs.

**Figure 5 ijms-21-01194-f005:**
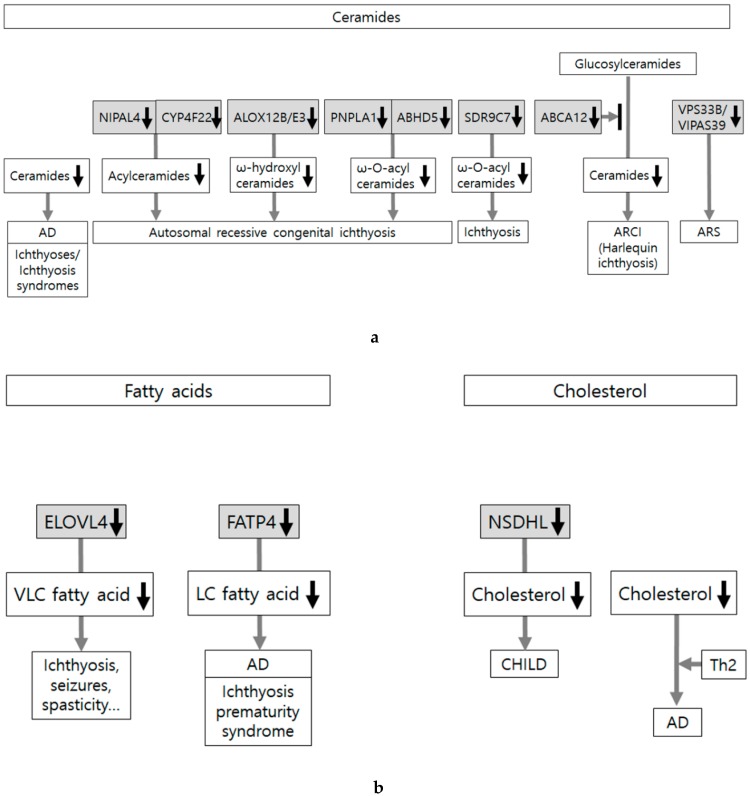
Skin lipid alteration in skin conditions showing epidermal barrier dysfunction. (**a**) Skin lipids are composed of ceramides, free fatty acids, and cholesterol. Abnormalities in lipid composition, transport, and extracellular organization induce abnormal lipid organization. Most of the reports are related to abnormalities in ceramides, which are associated with ichthyoses, ichthyosis syndromes, and atopic dermatitis (AD). Loss-of-function mutations in ABCA12 (ATP-binding cassette transporter A12) cause harlequin ichthyosis, the most severe phenotype of autosomal recessive congenital ichthyosis (ARCI). Loss-of-function mutations in NIPAL4 (NIPA like domain containing 4), CYP4F22, ALOX12B and ALOXE3, PNPLA1 (Patatin-like phospholipase domain-containing lipase 1), and ABHD5 (α/β hydrolase domain-containing protein 5) also cause ARCI. (**b**) Not much has been reported about the association between altered synthesis of free fatty acid/cholesterol and skin barrier dysfunction. Loss-of-function mutations in ELOVL4 (fatty acid elongase 4) and FATP4 (fatty acid transport protein 4) reduce very long-chain (VLC) or LC fatty acids and cause neurocutaneous disorder (characterized by ichthyosis, seizures, spasticity, intellectual disability, and ichthyosis) and ichthyosis prematurity syndrome, respectively. Although loss-of-function mutations in NSDHL (NADP-dependent steroid dehydrogenase-like) inhibit cholesterol synthesis and cause CHILD syndrome (Congenital Hemidysplasia with Ichthyosiform Erythroderma and Limb Defects), cholesterol depletion is not considered as adequate to induce atopic dermatitis-like alteration in the absence of Th2 cytokines.

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
