# Peer review of "Molecular Mechanism of Epidermal Barrier Dysfunction as Primary Abnormalities"

_ijms, 2020, doi:10.3390/ijms21041194_

Round 1

Reviewer 1 Report

In the work “Molecular Mechanism of Epidermal Barrier Dysfunction as Primary Abnormalities” the authors reviewed the influence of abnormalities in epidermal calcium gradients, filaggrin, cornified envelopes, desquamation of corneodesmosomes, and skin lipids in epidermal barrier dysfunction and epidermal barrier integrity.  The manuscript is well written, makes an extensive revision of the molecular mechanisms associated with epidermal barrier abnormalities, however there are some minor questions which should be addressed:

Pag 4 – improve Figure 1 for a better understanding Pag 5 – improve Figure 2

Author Response

Reviewer1

In the work “Molecular Mechanism of Epidermal Barrier Dysfunction as Primary Abnormalities” the authors reviewed the influence of abnormalities in epidermal calcium gradients, filaggrin, cornified envelopes, desquamation of corneodesmosomes, and skin lipids in epidermal barrier dysfunction and epidermal barrier integrity.  The manuscript is well written, makes an extensive revision of the molecular mechanisms associated with epidermal barrier abnormalities, however there are some minor questions which should be addressed:

Pag 4 – improve Figure 1 for a better understanding Pag 5 – improve Figure 2 

 Figure 1 is altered. Unnecessary portions are deleted in figure 2. Data from animal models are more convincing than those from cultured keratinocytes, therefore, references 23 and 24 for Figure 1 are changed.

Reviewer 2 Report

This is a detailed review of the possible role of changes in calcium gradients, keratin binding proteins, cornified envelope components, corneodesmosomes, and lipids, in damaging the epidermal barrier and causing skin disorders.

The review is well-referenced and the references are up to date. The authors have done a good job of covering relevant publications related to the topics they are covering. All in all, it's a useful paper from a literature review standpoint.

My concern is mainly focused on the writing style. This manuscript was hard to read, and although I understand the authors' first language is not English, the writing will have to be improved to be more "readable" before publication. For example, the authors use the word "could" throughout the text when the word "can" would be more appropriate in most cases. As an example, in the abstract the authors write that "Dysfunction of this barrier could cause skin disorders including..." There is no question that damage to the barrier CAN lead to skin disorders.

In addition, the sentence structure and length of the sentences often made it difficult to read. For example, in the abstract, one sentence reads; "Mechanisms linked to ichthyoses, atopic dermatitis without exacerbation or lesion, and early time of experimental irritation were included mainly in order to exclude the mechanism associated with epidermal barrier abnormalities resulted from preceding skin disorders".  Not only is the sentence structure incorrect, it is so long and confusing that readers will give up reading. 

The authors need to make sure the define abbreviations as soon as they are used in the text. For example, on line 124, the authors state that "The calcium gradients are also involved in CE rearrangement....". However, CE is not defined here but is only defined in line 181, a couple of pages later. Similarly, Tgm1, Tgm5, and SPRR are not defined at the time they are used in the text.

Finally, when describing results from published studies, it is not useful to say that the results showed that mutations in certain genes or proteins were "linked to", or "involved in " or "played a role" in a skin problem. Say exactly what the mutation caused. Did it make the skin problem worse or better? Although, the authors have done a pretty good job in some places of describing exactly what the studies they referenced showed, in other places they have not, and the reader is left trying to understand what effect a mutation, over-expression, etc. had on skin function. 

The importance of a review is to write it in a style that is easy for the reader to understand and follow, especially when a lot of data is being discussed.

Author Response

As for the recommendation of Reviewer 2;

My concern is mainly focused on the writing style. This manuscript was hard to read, and although I understand the authors' first language is not English, the writing will have to be improved to be more "readable" before publication. For example, the authors use the word "could" throughout the text when the word "can" would be more appropriate in most cases. As an example, in the abstract the authors write that "Dysfunction of this barrier could cause skin disorders including..." There is no question that damage to the barrier CAN lead to skin disorders.

<Could> was changed to <can> in this case. Other cases were also changed if there is no doubt on the contents.  

In addition, the sentence structure and length of the sentences often made it difficult to read. For example, in the abstract, one sentence reads; "Mechanisms linked to ichthyoses, atopic dermatitis without exacerbation or lesion, and early time of experimental irritation were included mainly in order to exclude the mechanism associated with epidermal barrier abnormalities resulted from preceding skin disorders".  Not only is the sentence structure incorrect, it is so long and confusing that readers will give up reading. 

The indicated sentence was divided into two sentences. The paper was revised according to your recommendation as much as possible.

The authors need to make sure the define abbreviations as soon as they are used in the text. For example, on line 124, the authors state that "The calcium gradients are also involved in CE rearrangement....". However, CE is not defined here but is only defined in line 181, a couple of pages later. Similarly, Tgm1, Tgm5, and SPRR are not defined at the time they are used in the text.

<Cornified envelopes> was firstly used in the Introduction, and abbreviations for this word <CEs> was moved to the right place. Small proline-rich proteins (SPRRs) was firstly shown in the front part of the <2.3. Cornified envelopes>. <Transglutaminase> was abbreviated as <TGase> as described in the front part of the <2.3. Cornified envelopes>. For Tgm1, <TGM1 encoding TGase 1 enzyme> was changed to <TGM1 (transglutaminase 1) encoding TGase1..>.

Finally, when describing results from published studies, it is not useful to say that the results showed that mutations in certain genes or proteins were "linked to", or "involved in " or "played a role" in a skin problem. Say exactly what the mutation caused. Did it make the skin problem worse or better? Although, the authors have done a pretty good job in some places of describing exactly what the studies they referenced showed, in other places they have not, and the reader is left trying to understand what effect a mutation, over-expression, etc. had on skin function. 

The paper was revised according to your recommendation as much as possible. The words <linked to, involved in, or played a role> in the contents concerning mutations in certain genes or proteins were changed.

This manuscript is a resubmission of an earlier submission. The following is a list of the peer review reports and author responses from that submission.